# The Calcium Channel Subunit Gamma-4 as a Novel Regulator of MafA in Pancreatic Beta-Cell Controls Glucose Homeostasis

**DOI:** 10.3390/biomedicines10040770

**Published:** 2022-03-25

**Authors:** Rui Wu, Alexandros Karagiannopoulos, Lena Eliasson, Erik Renström, Cheng Luan, Enming Zhang

**Affiliations:** Department of Clinical Sciences Malmö, Lund University Diabetes Centre, Lund University, 20502 Malmö, Sweden; rui.wu@med.lu.se (R.W.); alexandros.karagiannopoulos@med.lu.se (A.K.); lena.eliasson@med.lu.se (L.E.); erik.renstrom@med.lu.se (E.R.)

**Keywords:** beta-cell, calcium channel subunit gamma-4, *Cacng4*, IFG, IGT, MafA, CaMKII, prediabetes, diabetes, glucose metabolism, insulin

## Abstract

Impaired fasting glucose (IFG) and impaired glucose tolerance (IGT) are high-risk factors of diabetes development and may be caused by defective insulin secretion in pancreatic beta-cells. Glucose-stimulated insulin secretion is mediated by voltage-gated Ca^2+^ (Ca_V_) channels in which the gamma-4 subunit (Ca_V_γ4) is required for the beta-cell to maintain its differentiated state. We here aim to explore the involvement of Ca_V_γ4 in controlling glucose homeostasis by employing the *Ca_V_γ4*^−/−^ mice to study in vivo glucose-metabolism-related phenotypes and glucose-stimulated insulin secretion, and to investigate the underlying mechanisms. We show that *Ca_V_γ4*^−/−^ mice exhibit perturbed glucose homeostasis, including IFG and IGT. Glucose-stimulated insulin secretion is blunted in *Ca_V_γ4*^−/−^ mouse islets. Remarkably, Ca_V_γ4 deletion results in reduced expression of the transcription factor essential for beta-cell maturation, MafA, on both mRNA and protein levels in islets from human donors and *Ca_V_γ4*^−/−^ mice, as well as in INS-1 832/13 cells. Moreover, we prove that CaMKII is responsible for mediating this regulatory pathway linked between Ca_V_γ4 and MafA, which is further confirmed by human islet RNA-seq data. We demonstrate that Ca_V_γ4 is a key player in preserving normal blood glucose homeostasis, which sheds light on Ca_V_γ4 as a novel target for the treatment of prediabetes through correcting the impaired metabolic status.

## 1. Introduction

Type 2 diabetes (T2D) is progressive pathological hyperglycemia caused by insulin resistance and pancreatic beta-cell failure. Dysfunction of the beta-cells rapidly presents as damaged glucose-stimulated insulin secretion [1]. Development of T2D is inevitable when the deteriorated beta-cell function no longer copes with compensating for insulin resistance [2]. High risk for developing diabetes often relates to obesity, impaired fasting glucose (IFG) and impaired glucose tolerance (IGT) [3].

Maintaining the beta-cell identity is fundamental for preventing the development of diabetes [4]. The islet-enriched transcription factors control pancreatic beta-cell lineage determination, differentiation, and specification, e.g., MafA, Pdx1, MafB and Nkx6.1 [5,6]. MafA is exclusively expressed in adult beta-cells of man and rodents and plays an imperative role in promoting beta-cell maturation and function [5,7]. The unconditional presence of MafA for preserving the beta-cell identity and function has been broadly explored in different MafA-deleted animal models [8,9,10]. Such studies have demonstrated that MafA binds the insulin enhancer element, thus boosting insulin gene expression [11]. The importance of MafA is also supported by the fact that several vital islet transcription factors, such as Pdx1, Nkx6.1 and MafB, act in concert to stimulate MafA transcription [12,13]. In addition, Ca^2+^ signaling molecules, such as calcium/calmodulin-dependent protein kinase (CaMK) II, have been shown to control MafA expression in beta-cells as well [14,15]. However, whether extracellular stimuli such as Ca^2+^ can regulate MafA activity and further affect the beta-cell identity has not been explored.

The influx of extracellular Ca^2+^ in pancreatic beta-cells is mainly triggered by voltage-gated calcium (Ca_V_) channels. This process is initiated by glucose-evoked membrane depolarization that results in the discharge of docked insulin granules [16]. The beta-cell typically expresses several types of Ca_V_ channels. They are hetero-multimeric complexes composed of the transmembrane pore-forming alpha1 subunit and three auxiliary subunits: beta, alpha2delta, and gamma [17]. The predominant Ca_V_ alpha1 are well investigated and control beta-cell function through mediating Ca^2+^ entry [18,19,20], while the accessory beta and alpha2delta subunits have been identified as important regulators of Ca_V_ alpha1 trafficking [17,21]. However, the role of Ca_V_ gamma subunits (Ca_V_γ) in pancreatic beta-cells is only sketchily clarified. We have previously shown that Ca_V_ subunit gamma-4 (Ca_V_γ4) is necessary for preserving normal glucose-stimulated insulin secretion by permitting L-type Ca^2+^ currents/entry in both human and rodent pancreatic islets [22].

Ca_V_γ4 belongs to the transmembrane α-amino-3-hydroxy-5-methyl-4-isoxazolepropionic acid (AMPA) receptor regulatory protein (TARP) family, which modulates channel desensitization, deactivation, and trafficking, thereby affecting the channel decay kinetics and total charge transfer [23,24,25]. These findings were corroborated in Ca_V_γ4 (*Cacng4*) knockout mice that displayed disrupted excitatory postsynaptic currents in brain [26]. Furthermore, Ca_V_γ4 was shown to regulate both activation and inactivation of Ca_V_ channels via a direct association with the Ca_V_ alpha1 subunits [27,28]. In the beta-cell, it is noteworthy that among the eight Ca_V_γ variants, Ca_V_γ4 exhibits the highest expression in both man and mice [29], as well as in human beta-cell lines [30]. Intriguingly, Ca_V_γ4 levels are decreased in islets from T2D human donors, diabetic rodent models (Goto-Kakizaki (GK) rats and *db*/*db* mice), as well as in islets challenged with gluco-/lipotoxic environment [22]. Importantly, Ca_V_γ4 has been demonstrated as a direct downstream target of the beta-cell unique transcription factor MafA [22]. Nonetheless, the effect of Ca_V_γ4 on blood glucose levels and metabolic homeostasis in vivo remains unknown.

Here, we have extended our initial findings by employing the *Cacng4* overall knockout (*Cacng4^tm1Ran^*) mouse model with the aim of addressing the involvement of Ca_V_γ4 in glucose metabolism in vivo and underlying mechanism to provide a comprehensive view of the role of Ca_V_γ4 in the beta-cell. We found that genetic ablation of *Cacng4* in vivo confers IFG and IGT to the mice. Glucose-stimulated insulin secretion ex vivo is blunted in pancreatic islets from *Cacng4* knockout mice. This study provides novel insights into the causal signaling cascade from Ca_V_γ4 via CaMKII to MafA.

## 2. Materials and Methods

### 2.1. Human Islets

Human pancreatic islets were provided by EXODIAB Human Tissue Laboratory at Lund University through the Nordic Network for Clinical Islet Transplantation (Uppsala University, Uppsala, Sweden). All procedures were approved by the ethics committees at Uppsala and Lund Universities. The human islets were cultured in CMRL 1066 (ICN Biomedicals, Costa Mesa, CA, USA) supplemented with 10 mM HEPES, 2 mM L-glutamine, 50 μg/mL gentamicin, 0.25 μg/mL fungizone (Gibco, BRL, Gaithersburg, MD, USA), 20 μg/mL ciprofloxacin (Bayer Healthcare, Leverkusen, Germany) and 10 mM nicotinamide at 37 °C (5% CO_2_) for 1 to 5 days before arrival. The islets were then handpicked under the stereomicroscope. Islets from 6 non-diabetic donors were used in the experiment. Detailed information regarding human islet donors was listed in Appendix A.

### 2.2. Human Pancreatic Islet RNA-seq Data

Human islet RNA-seq data was acquired from a cohort of 188 human donors [31]. Pearson correlations between the expression levels of the genes of interest were performed on their log-transformed expression values.

### 2.3. Animal Studies

All animal experimentation was conducted in accord with accepted standards of humane animal care and approved by the local ethics committee.

#### 2.3.1. Animal Model

*Cacng4* knockout (*Cacng4^tm1Ran^*) mice were directly purchased from the Jackson Laboratory (JAX stock #028445), originally donated by Dr R. A. Nicoll, University of California, USA [26]. The live colony was maintained by mating the heterozygous (*Ca_V_γ4*^+/−^) mice together. Wild type (*Ca_V_γ4*^+/+^) mice inbred from the colony were used as control mice. Mice were housed in cages of 2–6, separated by sex. Some cages contained a mix of all three genotypes. Transgenic mice were identified by PCR of tail DNA using the following primers: Mutant Forward primer 5′-TAC CCG GTA GAA TTG ACC TGC-3′, Common primer 5′-ATC TGG TGA TGG CGT TCA GT-3′ and Wild type Forward primer 5′-GGA TCT ACA GCC GCA AGA AC-3′ with dreamtaq hot start DNA polymerase (#EP1702, Thermo Fisher Scientific, Waltham, MA, USA) under the following cycling conditions: 95 °C, 4 min; 95 °C, 30 s, 55 °C, 30 s, 72 °C, 30 s (35 cycles); 72 °C, 5 min. Reactions were separated on 2% agarose gels. The following band sizes were observed for Mutant = ~250 bp, Heterozygote = ~250 bp and 319 bp, Wild type = ~319 bp. Confirmation of *Cacng4* deletion was performed by qPCR (see Figure 1A).

#### 2.3.2. Blood Glucose Measurements

The mice underwent a 4-h fast each week from the time of weaning (3 weeks of age) for a period of 27 weeks (to 30 weeks of age) to examine the fasting blood glucose levels by using a OneTouch Lifescan (LifeScan Inc., Malvern, PA, USA) blood glucometer.

#### 2.3.3. IPGTT

Intraperitoneal glucose tolerance test (IPGTT) was performed in all 3-genotype mice without a fasting period from 11 to 15 weeks old. Blood glucose levels were measured with a handheld glucometer (OneTouch; Lifescan) prior to an intraperitoneal injection of 2 g glucose/kg body weight. Blood glucose values were quantified at 0, 5, 15, 30, 60, and 120 min post glucose injection.

### 2.4. Cell Culture

INS-1 832/13 cells (kindly donated by Dr C. B. Newgaard, Duke University, USA) were cultured in RPMI-1640 containing 11.1 mM D-glucose, and supplemented with 10% fetal bovine serum, 100 U/mL penicillin (Gibco, Gaithersburg, MD, USA), 100 μg/mL streptomycin (Gibco), 10 mM N-2 hydroxyethylpiperazine-N′-2-ethanesulfonic acid (HEPES), 2 mM glutamine, 1 mM sodium pyruvate, and 50 μM β-mercaptoethanol (Sigma), at 37 °C in a humidified atmosphere containing 95% air and 5% CO_2_.

### 2.5. Mouse Islet Isolation and Preparation

Intact primary mouse pancreatic islets were isolated by retrograde injection of a collagenase solution via the pancreatic duct from wild type and mutant mice and were handpicked under a stereomicroscope at room temperature. The isolated islets were kept in RPMI 1640 medium substituted with 10 mM D-glucose but lack of β-mercaptoethanol for culture. All indicated experiments were conducted on freshly isolated islets. No islet size and number alterations were observed in *Cacng4* knockout mice compared to wild type mice.

### 2.6. siRNA Transfection

INS-1 832/13 cells were seeded in a 24-well plate 1 day before transfection. 30 nM RNA interference oligonucleotides (Ambion, Austin, TX, USA) or Negative Control (Ambion, USA) were applied to silence target genes. Transfection reagent (Lipofectamine RNAiMAX, Invitrogen, Waltham, MA, USA) was used. For primary human islets, reverse transfection was performed to reach a faster and high-throughput transfection. Islets and siRNA-lipid complexes were prepared on the same day with Lipofectamine RNAiMAX (Invitrogen, USA). Transfection efficiency was measured by real-time PCR, western blotting, and visualized by BLOCK-iT Alexa Fluor Red Fluorescent Control (Invitrogen, USA) as described in the previous study [22].

### 2.7. Lentiviral Transfection

*Cacng4* or control (lentiviral particles without targeting any specific region) plasmids cloned in Lentiviral based shuttle vectors with mGFP tagged (ORIGENE, Rockville, MA, USA) were transformed into *E. coli* on LB-agar plates supplemented with 34 μg/mL chloramphenicol. Amplified plasmids were purified with QIAGEN Plasmid Midi Kit (QIAGEN, Hilden, Germany). Lentiviral vectors were amplified through transfection into HEK293T cells and harvested, followed by concentration and titration by service from Lund University Stem Cell Center Vector Unit. INS-1 832/13 cells were transfected with the *Cacng4* or control lentiviral vectors by directly adding into the culture medium under the calculation of 1 MOI (Multiplicity of Infection) for 72 h (change with fresh medium after 48 h transfection, and incubate another 24 h), followed by certain experiments as indicated in the Results session. Successful transfection was validated by qPCR, western blotting and visualized through UV light under microscopy as described in the previous study [22].

### 2.8. Insulin Secretion and Insulin Content Measurement

Freshly isolated mouse islets were preincubated with Krebs Ringer bicarbonate buffer (KRB), pH 7.4 (120 mM NaCl, 4.7 mM KCl, 2.5 mM CaCl_2_, 1.2 mM KH_2_PO_4_, 1.2 mM MgSO_4_, 25 mM NaHCO_3_, 10 mM HEPES and 1 mg/mL BSA) containing 2.8 mM glucose for 30 min at 37 °C with 95% O_2_ and 5% CO_2_ to obtain constant pH and oxygenation. The islets were then transferred to a 24-well plate with 8 size-matched islets in 0.5 mL KRB buffer supplemented with either 2.8 mM or 16.7 mM glucose per well and were incubated for 1 h at 37 °C with 95% O_2_ and 5% CO_2_. Immediately after incubation, an aliquot of the supernatant was collected for analysis of secreted insulin by mouse insulin ELISA kit (Mercodia, Uppsala, Sweden). The islets were then homogenized by RIPA buffer for 30 min at 4 °C followed by centrifugation (10,000× *g*, 15 min, 4 °C). The supernatant was collected for analysis of insulin content by mouse insulin ELISA kit (Mercodia, Sweden). Measurements were parallelly performed in 3–5 wells per condition.

For measuring insulin content in INS-1 823/13 cells, the cells were seeded in a 24-well plate followed by *Cacng4* siRNA or lentiviral transfection for 72 h. Then the insulin content measurements were performed under the same procedure as for mouse islets but detected by rat insulin ELISA kit (Mercodia, Sweden).

### 2.9. Real-Time Quantitative PCR

After transfection, total RNA from primary islets or INS-1 823/13 cells were extracted using the RNAeasy Kit (QIAGEN, Germany). 0.5–1 μg of RNA was used for cDNA synthesis. Primers of housekeeping genes HPRT1, B2M and POLR2A (Thermo Fisher Scientific, Waltham, MA, USA) and genes of interest (Thermo Fisher Scientific) which tagged FAM dyes, were used for amplification detection. The real-time PCR was carried out as follows: 50 °C for 2 min, 95 °C for 10 min, 40 cycles of 95 °C for 15 s, and 60 °C for 1 min by running on a ViiA 7 Real-Time System (Applied Biosystems, Waltham, MA, USA) with total reaction mixture (10 μL) consisting of 5 μL TaqMan Universal PCR Master Mix (Applied Biosystems), 2.5 μL 4× primer and 2.5 μL cDNA.

### 2.10. Western Blotting

INS-1 832/13 cells or mouse islets were homogenized in ice-cold RIPA buffer containing protease inhibitor (Roche, Basel, Switzerland) by shaking on ice for 30 min. The supernatant was collected after centrifugation (10,000× *g*, 15 min, 4 °C). Extracted total protein content was measured by Pierce BCA Protein Assay Kit (Thermo Fisher Scientific), and 10–20 μg of protein was electrophoresed on 4–15% SDS-PAGE (BIO-RAD, Hercules, CA, USA). The separated proteins were then transferred onto a PVDF membrane (BIO-RAD), followed by blocking with 5.0% fat-free dry milk in TBST (Tris-buffered saline with Tween 20) (pH 7.4; 0.15M NaCl, 10 mM Tris-HCl, and 0.1% Tween 20) for 1 h at room temperature. Afterwards, the membrane was incubated overnight at 4 °C with anti-Pdx1 (1:1000, Cell Signaling, Danvers, MA, USA), MafA (1:400, Abcam, Cambridge, United Kingdom), Nkx6.1 (1:1000, R&D Systems, Minneapolis, MN, USA), CaMKII (1:1000, Cell Signaling) and p-CaMKII (1:1000, Cell Signaling) antibodies followed by incubation with anti-rabbit IgG (1:2000, Cell Signaling) or goat anti-mouse IgG (1:2000, Dako, Glostrup, Denmark) at least 1 h at room temperature. Normalization was carried out by incubating the membrane with anti-PPIB (1:2000, Abcam) or Gapdh (1:1000, Abcam) antibodies. Immunoreactivity was detected using an enhanced chemiluminescence reaction (Pierce, Rockford, IL, USA) by SuperSignal West Femto Maximum Sensitivity Substrate (Thermo Fisher Scientific).

### 2.11. Statistics

The data are presented as means ± S.E.M. for the indicated number of observations or different experiments. The significance of random differences was analyzed by Student’s *t*-test. Pearson correlation coefficient (R) was used and tested (*t*-test) to measure the linear correlation between the expression of genes. One-way ANOVA with Tukey’s test and two-way ANOVA with Dunnett’s or Tukey’s or Two-stage linear step-up procedure of Benjamini, Krieger and Yekutieli’s test were used for multiple group comparisons. *p* < 0.05 was considered as significant. * or #, *p* < 0.05; **, *p* < 0.01; ***, *p* < 0.001.

## 3. Results

### 3.1. Cacng4 Knockout Mice Exhibit Impaired Fasting Glucose (IFG) and Impaired Glucose Tolerance (IGT)

We previously reported that Ca_V_γ4 is required for maintaining a functional differentiated beta-cell phenotype [22]. To investigate the more comprehensive role of Ca_V_γ4 in metabolism in vivo, the constitutive *Cacng4* knockout mouse (*Cacng4^tm1Ran^*, Jackson Laboratory), in which exon 3 is replaced with a neomycin resistance cassette, was used in the study [26]. First, successful deletion of *Cacng4* was confirmed by real-time qPCR using primers targeting exon 3 in isolated pancreatic islets from wild type (*Ca_V_γ4*^+/+^), heterozygous (*Ca_V_γ4*^+/−^), and homozygous (*Ca_V_γ4*^−/−^) *Cacng4* knockout mice. Ca_V_γ4 mRNA expression was downregulated by almost 50% in *Ca_V_γ4*^+/−^ mice, and a further complete abrogation of Ca_V_γ4 was verified in *Ca_V_γ4*^−/−^ mice (Figure 1A).

We then examined the effect of Ca_V_γ4 ablation on blood glucose levels. Interestingly, an elevated 4-h fasting blood glucose level was observed in both male *Ca_V_γ4*^+/−^ and male *Ca_V_γ4*^−/−^ mice at the age of 15–19 weeks compared to the male *Ca_V_γ4*^+/+^ mice (Figure 1B), but not in female transgenic mice (Figure 1C). To corroborate further the impact of Ca_V_γ4 on glucose metabolism in vivo, we performed the intraperitoneal glucose tolerance test (IPGTT). Notably, in contrast to *Ca_V_γ4*^+/+^ mice, administration of glucose without fasting led to a significantly increased blood glucose level at 15 min post-injection in male *Ca_V_γ4*^−/−^ mice at 11–15 weeks of age (Figure 1D). The raised blood glucose level was also observed at 5 min post-injection in female *Ca_V_γ4*^−/−^ mice at the same age (Figure 1E). These results show that Ca_V_γ4 is involved in the regulation of blood glucose metabolism in an age- and gender-specific manner.


Figure 1Impaired fasting blood glucose and glucose tolerance in *Cacng4* knockout mice: (**A**) Ca_V_γ4 (*Cacng4*) mRNA levels in isolated islets from wild type (*Ca_V_γ4*^+/+^), heterozygous (*Ca_V_γ4*^+/−^), and homozygous (*Ca_V_γ4*^−/−^) *Cacng4* knockout mice, *n* = 3, 4 and 3 mice, respectively. *** *p* < 0.001 (*Ca_V_γ4*^+/+^ vs. *Ca_V_γ4*^−/−^) and ** *p* < 0.01 (*Ca_V_γ4*^+/+^ vs. *Ca_V_γ4*^+/−^ and *Ca_V_γ4*^+/−^ vs. *Ca_V_γ4*^−/−^). (**B**) Blood glucose levels in 4-h fasted male *Ca_V_γ4*^+/+^, *Ca_V_γ4*^+/−^ and *Ca_V_γ4*^−/−^ mice from 3 weeks to 30 weeks. *n* = 5, 21 and 3 mice (≤ 6 weeks); 9, 9 and 5 mice (7–10 weeks); 8, 19 and 5 mice (11–14 weeks); 7, 10 and 4 mice (15–19 weeks, ** *p* = 0.003 (*Ca_V_γ4*^+/+^ vs. *Ca_V_γ4*^+/−^), * *p* = 0.034 (*Ca_V_γ4*^+/+^ vs. *Ca_V_γ4*^−/−^)); 4, 27 and 4 mice (≥20 weeks), respectively. (**C**) As in (**B**) but in female *Ca_V_γ4*^+/+^, *Ca_V_γ4*^+/−^ and *Ca_V_γ4*^−/−^ mice. *n* = 8, 20 and 4 mice (≤ 6 weeks); 6, 19 and 15 mice (7–10 weeks); 6, 21 and 13 mice (11–14 weeks); 6, 14 and 14 mice (15–19 weeks); 8, 14 and 3 mice (≥20 weeks), respectively. (**D**) Plasma glucose concentrations during IPGTT (2 g/kg) in male *Ca_V_γ4*^+/+^, *Ca_V_γ4*^+/−^ and *Ca_V_γ4*^−/−^ mice at 11–15 weeks of age. *n* = 7, 9 and 5 mice, respectively. * *p* = 0.025, *Ca_V_γ4*^+/+^ vs. *Ca_V_γ4*^−/−^ at 15 min. (**E**) As in (**D**) but in female mice. *n* = 10, (*Ca_V_γ4*^+/+^), 14 (*Ca_V_γ4*^+/−^) and 14 (*Ca_V_γ4*^−/−^) mice, respectively. * *p* = 0.024, *Ca_V_γ4*^+/+^ vs. *Ca_V_γ4*^−/−^ at 5 min. Data is presented as Mean ± SEM and were analyzed with one-way ANOVA with Tukey’s multiple comparisons test (**A**); and two-way ANOVA with Tukey’s, Dunnett’s or Two-stage linear step-up procedure of Benjamini, Krieger and Yekutieli’s multiple comparisons tests (**B**–**E**).
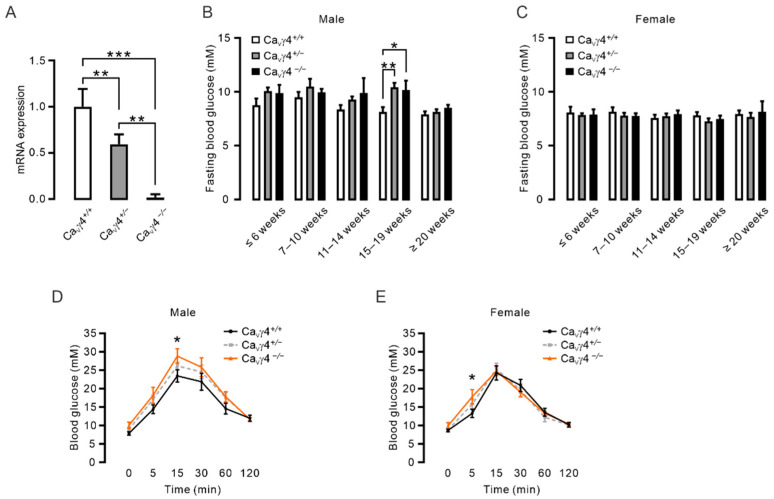



### 3.2. Ca_V_γ4 Is Required for Glucose-Stimulated Insulin Secretion in Beta-Cells

The observation of Ca_V_γ4-related effects on glucose homeostasis in vivo may result from damaged beta-cell function [22]. To explore this possibility, ex vivo glucose-stimulated insulin secretion assay was conducted in pancreatic islets from *Cacng4* knockout mice at 11–15 weeks of age. As expected, in freshly isolated *Ca_V_γ4*^−/−^ mouse islets, glucose-stimulated insulin secretion was markedly reduced in comparison to that in *Ca_V_γ4*^+/+^ mouse islets (Figure 2A). A tendency of attenuated glucose-stimulated insulin secretion was seen in Ca_V_γ4 heterozygotes islets (Figure 2A). This data is in line with our previous findings [22], demonstrating that Ca_V_γ4 is of importance for glucose-stimulated insulin secretion in pancreatic beta-cells, and in turn, involved in the regulation of global glucose metabolism.

Given the decisive role of Ca_V_γ4 in mediating pancreatic beta-cell function, the underlying molecular mechanism is of great interest. It has been documented that loss of Ca_V_γ4 induces beta-cell dedifferentiation [22], which is attributed to the decreased expression of critical beta-cell identity transcription factors (e.g., MafA, Pdx1, Nkx6.1) during glucotoxicity [4,32]. Accordingly, we assessed mRNA levels of key islet-enriched genes as well as the beta-cell dedifferentiation marker aldehyde dehydrogenase1a3 (Aldh1a3) [33] in *Cacng4* knockout mouse islets. Real-time qPCR revealed that *MafA* mRNA was significantly low in islets from both *Ca_V_γ4*^+/−^ and *Ca_V_γ4*^−/−^ mice compared with islets from wild type mice (Figure 2B). Remarkably, *Aldh1a3* mRNA expression was ~35% higher in *Ca_V_γ4*^−/−^ islets than that in wild type islets, which agrees with our earlier finding that Ca_V_γ4 deletion leads to loss of beta-cell identity by dedifferentiation [22] (Figure 2B). Collectively, these data support the hypothesis that Ca_V_γ4-ablation-induced disturbed insulin secretion results from downregulation of beta-cell identity genes (i.e., *MafA*), leading to beta-cell dedifferentiation.

### 3.3. Ca_V_γ4 Regulates MafA via CaMKII

The gene expression data in transgenic mouse islets (Figure 2B), were next complemented by assaying mRNA and protein levels of MafA and Pdx1 in human islets and INS-1 832/13 cells when silencing or overexpressing Ca_V_γ4. Notably, *MAFA* mRNA expression, but not *PDX1*, was significantly decreased in Ca_V_γ4-silenced non-diabetic human islets (Figure 3A). This was supported by data obtained from western blotting (Figure 3B). Confirmation in INS-1 832/13 cells by lentiviral-based overexpressing Ca_V_γ4 also showed that *MafA* mRNA was substantially enhanced, whereas *Pdx1* mRNA remained unaltered (Figure 3C). These data together provide evidence for MafA being one of the downstream targets of Ca_V_γ4 in regulating beta-cell function and differentiation status. Subsequently, we focused on how Ca_V_γ4 exerts its effects on MafA in beta-cells.

It is well established that CaMKII inhibition causes MafA downregulation in pancreatic beta-cells [14,15]. Moreover, as Ca_V_γ4 affects intracellular calcium homeostasis [22], it is likely that Ca_V_γ4 may regulate CaMKII expression/activation. This leads to the exciting possibility that CaMKII is the missing link in the chain of events from Ca_V_γ4 ablation to reduced MafA levels. Indeed, silencing Ca_V_γ4 strongly suppressed the expression of both CaMKII and phosphorylated-CaMKII in INS-1 832/13 cells (Figure 3D). The decreased extent of total CaMKII in Ca_V_γ4-silenced beta-cells was similar to that of phosphorylated-CaMKII, suggesting no additional regulation of Ca_V_γ4 on CaMKII phosphorylation (Figure 3E). This reduced CaMKII protein expression was ideally confirmed in *Ca_V_γ4*^−/−^ mouse islets in comparison to wild type islets (Figure 3F). To corroborate the relationship between Ca_V_γ4 and CaMK gene family, RNA-seq data of pancreatic islets from 188 human donors was analyzed. In agreement with our hypothesis, Ca_V_γ4 (*CACNG4*) expression robustly exhibited a positive correlation with all 4 CaMKII subunits in human islets (Figure 3G). In addition, a similar correlation was also observed between CaMKII variants and *MAFA* (Figure 3H).

We finally studied the outcome of the chain of events that Ca_V_γ4 through mediating CaMKII levels controls MafA expression, which in turn affects insulin production. As expected, removal of Ca_V_γ4 resulted in a decline in insulin content in INS-1 832/13 cells (Figure 4A). Conversely, overexpression of Ca_V_γ4 increased the total intracellular insulin content in INS-1 832/13 cells (Figure 4B). These findings were strongly supported by the positive correlation of mRNA expressions between Ca_V_γ4 (*CACNG4*) and insulin (*INS*) in human pancreatic islets (Figure 4C).

Taken together, these results demonstrate that reduction of Ca_V_γ4 in the beta-cell decreases MafA expression by suppressing CaMKII expression and activation, which reduces insulin gene expression and secretion and loss of the mature beta-cell identity (Figure 4D).

## 4. Discussion

Studies in human and rodent pancreatic islets in vitro have established that Ca_V_γ4 determines beta-cell function by enhancing Ca^2+^ entry and the subsequent glucose-stimulated insulin secretion [22]. Here, we further demonstrate that Ca_V_γ4 is essential for in vivo glucose homeostasis in mice. Our results disclose that Ca_V_γ4 is pivotal for maintaining normal fasting blood glucose levels and glucose tolerance in vivo by regulating the beta-cell determining transcription factor MafA expression. When this signaling pathway is disrupted in Ca_V_γ4-ablated beta-cells, glucose metabolism is impaired and insulin-producing and -releasing properties of beta-cells are deteriorated (Figure 4D).

### 4.1. Ca_V_γ4 and Glucose Metabolism

Ca_V_γ subunits have been reported to be expressed in both fetal and adult human brains and are widespread in different tissues [34,35]. All eight variants are detected in human pancreatic islets with, however, only Ca_V_γ4 being differentially expressed in islets from T2D human donors and diabetic animal modes, GK rats and *db*/*db* mice [22], rendering the rationale for aiming at Ca_V_γ4. In fact, in vitro data in human islets identify that the changes in Ca_V_γ4 expression, with resultant effects on beta-cell stimulus-secretion coupling, can provide a novel interpretation for beta-cell failure [22]. Accordingly, Ca_V_γ4 plays key roles in the brain. There is evidence that Ca_V_γ4 rescues the defective excitatory synaptic transmission in cerebellar granule cells from the *Cacng2* mutant mice (*stargazer* mice) [23]. This is confirmed by the discovery that Ca_V_γ4 silencing is seizure inducible [36]. Importantly, the total charge transfer through postsynaptic AMPA receptors at central synapses is diminished in *Cacng4* knockout mice [26]. These studies together strongly imply that Ca_V_γ4 may exert an indispensable effect in modulating metabolic process and glucose homeostasis in vivo. Indeed, the ability to control fasting blood glucose levels was lost in male *Cacng4* knockout mice (Figure 1B). Interestingly, this was unaffected in female *Cacng4* transgenic mice (Figure 1C). We speculate that this gender-specific phenotype may associate with estrogens signal networks in brain. The physiology of estrogen action on managing glucose transport into cells and glycolysis primarily through its receptors have been intensively studied. Withdrawal of estrogens or estrogen receptors causes dysregulation of glucose metabolism and body weight and is associated with development of diabetes [37]. Given the significant role of Ca_V_γ4 in neurons, it would not be surprising that the estrogen physiology is disturbed by Ca_V_γ4 deletion in brain. Future investigations focused on Ca_V_γ4-derived estrogen signal regulation is warranted. Secondly, this elevated 4-h fasting blood glucose levels and IGT seen in *Cacng4* knockout mice, jointly with ex vivo downregulated insulin secretion in *Ca_V_γ4*^−/−^ mouse islets (Figure 1B,D,E and Figure 2A), perfectly confirmed our hypothesis that Ca_V_γ4 governs glucose homeostasis in vivo. Be that as it may, the effects of Ca_V_γ4 on plasma insulin levels and insulin responsive peripheral tissues are unclear, since insulin resistance is highly related to prediabetes [2] which is marked by IFG and IGT. Additionally, both IFG and IGT became manifest in *Cacng4* knockout mice (Figure 1B,D,E). Therefore, the plasma insulin concentration and insulin action may also be lowered in these mice that partly contributes to the impaired glucose homeostasis in vivo. The insulin levels in plasma and insulin signaling transduction via insulin receptors in fat, muscle, and liver cells from *Cacng4* knockout mice will be examined in our next paper. In agreement with our previous findings [22], *Ca_V_γ4*^−/−^ mouse islets exhibited blunted glucose-stimulated insulin secretion and upregulated expression of *Aldh1a3*, a beta-cell dedifferentiation marker [33] (Figure 2A,B), suggesting the vital role of Ca_V_γ4 in determining beta-cell functionality. In summary, Ca_V_γ4 is more important than anticipated in controlling metabolism in vivo and appears a key character in the regulation of blood glucose levels and insulin secretion.

### 4.2. Ca_V_γ4 and MafA Signaling

Apart from allowing Ca^2+^ entry to activate Ca^2+^-dependent signaling pathways, Ca_V_ channels have been increasingly documented that are substantial for modification of gene expression and beta-cell differentiation [18,22,38]. In this respect, they behave directly as transcription factors (by Ca_V_ subunits or fragments) or indirectly by Ca^2+^-binding/-activated proteins to regulate a plethora of downstream transcription factors, for instance, CREB, NFAT, DREAM and MafA [14,15,38]. Being the principal beta-cell-enriched transcription factor, MafA is crucial for beta-cell maturation and mediating insulin transcription and other genes in a glucose-responsive manner [5,39]. Our previous study revealed that Ca_V_γ4 expression is under the control of MafA [22]. On the other hand, as part of Ca_V_ channels, Ca_V_γ4 is also responsible for Ca^2+^ entry and transients in the cytosol [22], raising the likelihood of a negative feedback from Ca_V_γ4 to MafA. Indeed, we here add new knowledge to the effect that Ca_V_γ4 has a profound impact on MafA expression, and most importantly, this negative feedback is conducted via the Ca^2+^-binding enzyme CaMKII. First, MafA expression was reduced in *Cacng4* knockout mouse islets, and its levels in human islets and INS-1 832/13 cells can be manipulated by silencing or overexpressing Ca_V_γ4 (Figure 2B and Figure 3A–C). Then, CaMKII activation has been shown to clearly modulate MafA expression and insulin secretion in pancreatic beta-cells [14,15]. This conclusion is extended by our western blotting data that both CaMKII and phosphorylated-CaMKII expressions were weakened in Ca_V_γ4-silenced beta-cells (Figure 3D), and the decreased CaMKII protein levels were also perfectly observed in islets from *Ca_V_γ4*^−/−^ mice (Figure 3F). Conspicuously, the positive associations between CaMKII subunits and Ca_V_γ4 (*CACNG4*) as well as *MAFA* were firmly echoed by the analysis of human pancreatic islet RNA-seq data (Figure 3G,H). However, the mechanism behind that bridges Ca_V_γ4 and CaMKII is unexplored and is of interest to study in the future. We would speculate this regulatory interaction is either resulted from the intracellular Ca^2+^ concentrations determined by Ca_V_γ4 expression [22], or simply through a specific binding [40]. Finally, insulin content was significantly associated with Ca_V_γ4 levels in both human islets and INS-1 832/13 cells (Figure 4A–C), which fully agrees with the fact that MafA is a dedicated activator of the insulin transcription [7].

### 4.3. Limitations of the Study

The principal limitations of the present study are the inadequate experiments in the *Cacng4* knockout mice. Given the observed prediabetes associated phenotypes (IFG and IGT), first and foremost, in-depth in vivo exploration of the involvement of Ca_V_γ4 in insulin resistance need to be conducted. It would be essential to examine the influences of Ca_V_γ4 on plasma insulin levels and insulin signaling pathways in insulin target peripheral tissues, for example liver, fat, and muscle. The reciprocal regulation between Ca_V_γ4 and estrogen, which might be responsible for the gender-specific phenotype, requires further in vivo study, and the employment of beta-cell conditional knockout mice would be necessary. In addition, it would be important to investigate the regulatory machinery of Ca_V_γ4 on CaMKII.

## 5. Conclusions

We identify a comprehensive role of the Ca_V_γ4 subunit for regulating glucose homeostasis in mice in vivo. Our results suggest new beneficial effects of Ca_V_γ4 with the potential to preclude the development of prediabetes-related IFG and IGT through its influence on insulin secretion capability in islets. This fairly resonates with the existing view that Ca_V_γ4 contributes the key impingement to sustaining a functional beta-cell phenotype [22]. Strikingly, we clarify a novel negative feedback pathway from Ca_V_γ4 to MafA by the involvement of CaMKII. These results together shed light on prediabetes treatment in the future by aiming at a novel target, the Ca_V_γ4 subunit, to prevent hyperglycemia.

## Figures and Tables

**Figure 2 biomedicines-10-00770-f002:**
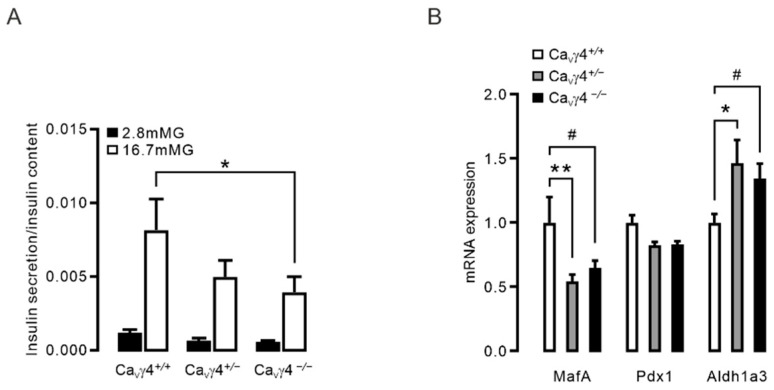
Insulin secretion and gene expression in *Cacng4* knockout mouse islets: (**A**) Glucose-stimulated insulin secretion in *Ca_V_γ4*^+/+^, *Ca_V_γ4*^+/−^ and *Ca_V_γ4*^−/−^ mouse islets. *n* = 6, 7 and 6 mice, respectively. * *p* = 0.016. (**B**) *MafA*, *Pdx1* and *Aldh1a3* mRNA expression in *Ca_V_γ4*^+/+^, *Ca_V_γ4*^+/−^ and *Ca_V_γ4*^−/−^ mouse islets. *n* = 4, 4 and 4 mice, respectively. ** *p* < 0.01, * *p* < 0.05 and ^#^
*p* < 0.05. Data are presented as Mean ± SEM and were analyzed with two-way ANOVA with Dunnett’s multiple comparisons test.

**Figure 3 biomedicines-10-00770-f003:**
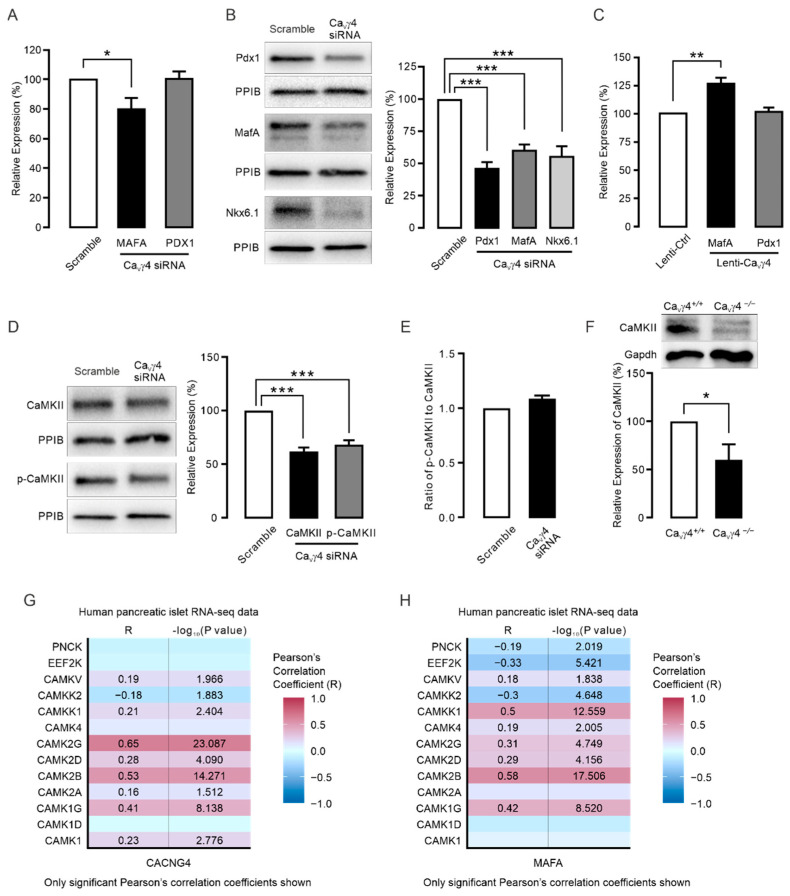
CaMKII expression is downregulated in Ca_V_γ4-silenced beta-cells: (**A**) *PDX1* and *MAFA* mRNA expression in Ca_V_γ4-silenced human islets. *n* = 4 (*PDX1*) and 6 (*MAFA*) donors. * *p* = 0.031. (**B**) Pdx1, MafA and Nkx6.1 immunoblotting and means of expression in Ca_V_γ4-silenced INS-1 832/13 cells. *n* = 4, *** *p* < 0.001. (**C**) As in (**A**) but in Ca_V_γ4-overexpressed INS-1 832/13 cells. *n* = 4, ** *p* = 0.002. (**D**) Decreased CaMKII and phosphorylated-CaMKII expressions in Ca_V_γ4-silenced INS-1 832/13 cells. *n* = 4, *** *p* < 0.001 for both. (**E**) Ratio of protein expressions in (**D**) of phosphorylated-CaMKII to CaMKII. *n* = 4, *p* > 0.05. (**F**) CaMKII protein levels in *Ca_V_γ4*^+/+^ and *Ca_V_γ4*^−/−^ mouse islets. *n* = 4 mice each. * *p* < 0.05. (**G**) Pearson’s correlation coefficients (R) calculated by mRNA expression (RNA-seq, human islets) between the CAMK family genes and *CACNG4*. *n* = 188 human donors. (**H**) As in (**G**) but between CAMK family genes and *MAFA*. *n* = 188 human donors. Data are presented as Mean ± SEM and were analyzed with one-way ANOVA with Dunnett’s multiple comparisons test (**A**–**D**); and two-tailed Student’s *t*-test (**E**–**H**). See Appendix A for the details of the human donors utilized for the experiment in this study.

**Figure 4 biomedicines-10-00770-f004:**
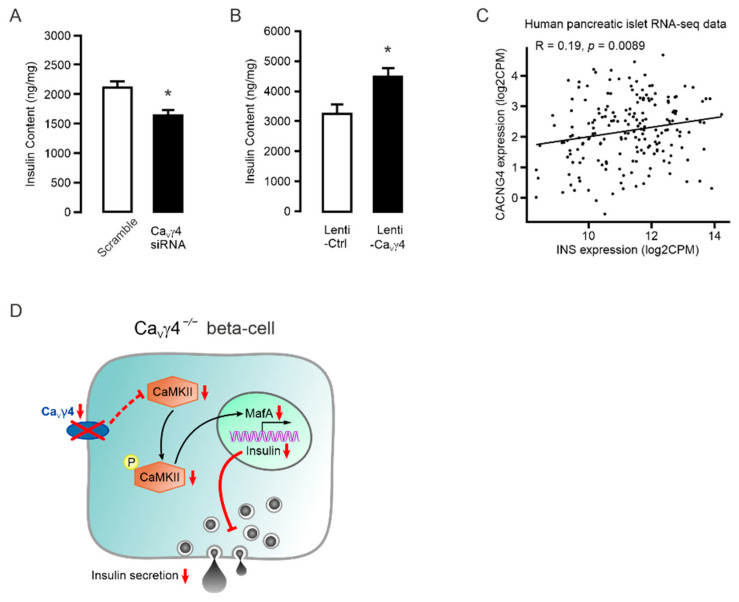
Ca_V_γ4 regulates insulin production in beta-cells: (**A**) Reduced insulin content in Ca_V_γ4-silenced INS-1 832/13 cells. *n* = 4, * *p* = 0.014. (**B**) Increased insulin content in Ca_V_γ4-overexpressed INS-1 832/13 cells. *n* = 4, * *p* = 0.035. (**C**) Pearson correlation performed between the log2-transformed normalized expression values of *CACNG4* and *INS* in Counts Per Million (CPM). *n* = 188 human donors. (**D**) Schematic model of cascading effects from Ca_V_γ4 deletion through CaMKII expression to insulin secretion in *Ca_V_γ4*^−/−^ beta-cell. See main text for details. Data are presented as Mean ± SEM and were analyzed with two-tailed unpaired Student’s *t*-test.

## Data Availability

All data generated and/or analyzed during the current study are available from the corresponding authors on reasonable request.

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
