# Peer review of "The Calcium Channel Subunit Gamma-4 as a Novel Regulator of MafA in Pancreatic Beta-Cell Controls Glucose Homeostasis"

_biomedicines, 2022, doi:10.3390/biomedicines10040770_

Round 1
Reviewer 1 Report
The reviewer acknowledges the authors of the manuscript entitled "The calcium channel subunit gamma-4 as a novel regulator of MafA in pancreatic beta-cell controls glucose homeostasis". The manuscript is well written and constructed and the thematic is relevant for the field of diabetes treatment. However the reviewer has some comments/suggestions to the authors:
- throughout the manuscript the authors state that the results from this manuscript support that calcium channel subunit gamma-4 can be a novel target for the treatment of prediabetes since modulates beta-cell function. However, the reviewer would like that authors clarify those sentences since in prediabetes we still have a functional beta-cell function since the major dysfunction is insulin resistance in many tissues.
- since in figure 1 authors show that males have a greater impact of the KO of the channel, the reviewer would like to know if the mouse islets isolated and described in 2.5 section of the methods are from male or female mice. If both genders were included a special attention should be taken to the conclusions.
- in figure 3 the reviewer would like to see the representation of the ratio pCaMKII/CaMKII since it looks that there is a decrease in the total expression of this protein but the phosphorylation it seems to be similar or even increased. If that is the case a different conclusion should be made about the inactivation of the CaMKII signaling pathway.
Reviewer 2 Report
Authors succeeded in finding of unprecedented phenotype in Cavγ4KO mouse. It is very interesting in the point: Increased glycemia in fasting state and glucose-loaded state in 15-19 week-old KO mice are prominent in male mice, suggesting the involvement of sex hormones in Cavγ4-mediated regulation of glycemia. I am also interested in the author's scheme which Cavγ4 maintains CaMKII-MafA-mediated synthesis of insulin. However, this author's scheme is not sufficiently supported by the results. I think that this paper should be improved or reconsider/rebuild the logic before publication, especially in the points listed below.
- Fasting blood glucose level in 15-19-week-old male Cavγ4KO(-/-) mice should be high with statistical significance (fig1B; p=0.06).
- (Related to 1) If this elevation in blood glucose level can be reliable, why the change is seen in this narrow time window? Why is this not seen in younger male mice and recovered in older male mice?
- Fig2B. Why MafA mRNA level is not reduced in Cavγ4-/- mice? Is it really OK to conclude that "Cavγ4 gene reduction cause MafA downregulation"?
- Related to 3. SEM error bar in Cavγ4+/+ in MafA column is very long, indicates that there are large deviations in the data mass of MafA mRNA in Cavγ4+/+ mice. In general, it seems to be difficult to get statistical significance between Cavγ4+/+ and Cavγ4+/- in MafA column. I recommend to re-perform statistical analysis.
- Cavγ4KO mouse in this work is produced by the technique of conventional knockout. Is there any developmental effect of KO? How about the morphology, size and the number of islets or beta-cells in Cavγ4KO mouse? This point should be clarified or discussed.
- Results in Fig4B should be performed using islets or β-cells from Cavγ4-/- mice. Off course I know that lentiviral transduction requires at least 3-4 days for waiting for sufficient expression, so isolated islets or beta-cells may lose their original properties of in vivo β-cell. However, even such situation, more reliable results would be taken than INS-1 cells.
- Related to 6. CaMKII downregulation in β-cell in Cavγ4KO mice should be shown.
- I would like to know whether the Ca2+ signaling, membrane potentials and number of insulin content in each β-cell is different among Cavγ4+/+, +/- and -/-.
Reviewer 3 Report
Wu et al. aimed to explore if CaVγ4 contributes to controlling glucose homeostasis and underlying mechanisms. CaVγ4-/- mice were used to investigate in vivo glucose-metabolism-related phenotypes and glucose-stimulated insulin secretion. They show that CaVγ4-/- mice exhibit perturbed glucose homeostasis, includingIFG and IGT.
The study covers some issues that have been overlooked in other similar topics. The structure of the manuscript appears adequate and well divided in the sections. Moreover, the study is easy to follow, but few issues should be improved. Some of the comments that would improve the overall quality of the study are:
1-) Abstract and Introduction section: Please better describe the aim of the work;
2-) The manuscript needs grammar correction. Please also check typos thorough the text;
3-) Limitations of the study needs to be added;
4-) Conclusion Section: This paragraph is missing (only partially reported in 4.2 sub-section). Please add it.
Round 2
Reviewer 2 Report
All of my questions and concerns are disappeared in the response letter and current version of manuscript. I'm looking forward to the author's next work.